# Rats Orally Administered with Ethyl Alcohol for a Prolonged Time Show Histopathology of the Epididymis and Seminal Vesicle Together with Changes in the Luminal Metabolite Composition

**DOI:** 10.3390/biomedicines12051010

**Published:** 2024-05-03

**Authors:** Chayakorn Taoto, Nareelak Tangsrisakda, Wipawee Thukhammee, Jutarop Phetcharaburanin, Sitthichai Iamsaard, Nongnuj Tanphaichitr

**Affiliations:** 1Department of Anatomy, Faculty of Medicine, Khon Kaen University, Khon Kaen 40002, Thailand; chayakorntaoto@kkumail.com (C.T.); nareelak@kku.ac.th (N.T.); 2Research Institute for Human High Performance and Health Promotion, Khon Kaen University, Khon Kaen 40002, Thailand; wipath@kku.ac.th; 3Department of Systems Biosciences and Computational Medicine, Faculty of Medicine, Khon Kaen University, Khon Kaen 40002, Thailand; jutarop@kku.ac.th; 4Khon Kaen University Phenome Centre, Khon Kaen University, Khon Kaen 40002, Thailand; 5Chronic Disease Program, Ottawa Hospital Research Institute, Ottawa, ON K1Y 4E9, Canada; 6Department of Obstetrics and Gynecology, Faculty of Medicine, University of Ottawa, Ottawa, ON K1Y 8L6, Canada

**Keywords:** ethyl alcohol, epididymis, seminal vesicle, histopathology, lipid peroxidation, apoptosis, epididymal fluid, seminal vesicle fluid, metabolomics, proton nuclear magnetic resonance

## Abstract

Prolonged ethanol (EtOH) consumption is associated with male infertility, with a decreased spermatogenesis rate as one cause. The defective maturation and development of sperm during their storage in the cauda epididymis and transit in the seminal vesicle can be another cause, possibly occurring before the drastic spermatogenesis disruption. Herein, we demonstrated that the cauda epididymis and seminal vesicle of rats, orally administered with EtOH under a regimen in which spermatogenesis was still ongoing, showed histological damage, including lesions, a decreased height of the epithelial cells and increased collagen fibers in the muscle layer, which implicated fibrosis. Lipid peroxidation (shown by malondialdehyde (MDA) levels) was observed, indicating that reactive oxygen species (ROS) were produced along with acetaldehyde during EtOH metabolism by CYP2E1. MDA, acetaldehyde and other lipid peroxidation products could further damage cellular components of the cauda epididymis and seminal vesicle, and this was supported by increased apoptosis (shown by a TUNEL assay and caspase 9/caspase 3 expression) in these two tissues of EtOH-treated rats. Consequently, the functionality of the cauda epididymis and seminal vesicle in EtOH-treated rats was impaired, as demonstrated by a decreases in ^1^H NMR-analyzed metabolites (e.g., carnitine, fructose), which were important for sperm development, metabolism and survival in their lumen.

## 1. Introduction

Alcoholic beverages containing ethanol (ethyl alcohol, EtOH) are legally consumed by adults in most countries. Chronic EtOH consumption is associated with many diseases including liver cirrhosis, pancreatitis, memory impairment, colorectal cancer, hypertensive heart disease and infertility [1]. Particularly, EtOH adversely affects histology and functions of male reproductive tissues in both men and experimental animals. In men drinking alcohol for a long time, their sex hormonal levels such as follicle stimulating hormone (FSH), luteinizing hormone (LH) and testosterone are decreased [2]. The same decreases have also been observed in animal models [3,4]. Histopathology in the seminiferous tubules has been reported in patients and animals with prolonged EtOH consumption. This includes the depletion and dissociation of testicular germ cells in the seminiferous tubule epithelium together with their sloughing into the lumen. These abnormalities result in decreases in the tubule diameter [5,6,7,8]. Apoptosis is also increased in testicular germ cells [7,8], and lipid accumulation at the base of Sertoli cells is noted in EtOH-consuming rats [9]. All of this histopathology is closely associated with a decreased rate of spermatogenesis [5,6,10], and in some men with heavy EtOH consumption, they could have the Sertoli cell-only syndrome (i.e., Sertoli cells are the main cells in the seminiferous tubules [6]).

The impairment observed in the testis of males with prolonged EtOH consumption could be from decreased levels of FSH, LH and testosterone. However, EtOH is also metabolized into harmful agents. Acetaldehyde is generated from EtOH through the activity of aldehyde dehydrogenase or CYP2E1 [11], both present in the testis [12,13]. In fact, CYP2E1 expression is induced by EtOH, and its catalysis of EtOH generates not only acetaldehyde but also reactive oxygen species (ROS) [14], which in turn causes the peroxidation of polyunsaturated fatty acids (PUFA) [11]. This lipid peroxidation results in the production of lipid aldehydes such as malondialdehyde (MDA) and other lipid peroxidation products such as hydroxynoenol (HNE). Acetaldehyde, lipid aldehydes (e.g., MDA) and lipid enols (e.g., HNE) are very reactive compounds. They form adducts with DNA and proteins, and this could lead to cell apoptosis [11,15]. In fact, rats, which consume EtOH for a prolonged period, showed increased levels of MDA in the testis [8,16] and a higher degree of testicular germ cell apoptosis [7,8,16,17,18,19].

In certain studies, spermatogenesis is still ongoing in rats prolongedly consuming EtOH. A close to normal organization of testicular germ cells and Sertoli cells in the seminiferous tubules is described in these EtOH rats, although a slight decrease in sperm production is noted. However, only 50% of caudal epididymal sperm are viable, and a number of them have an abnormal morphology, especially in the head region [20,21,22]. The much less severe disruption of spermatogenesis in these studies could be due to the lower doses of EtOH in the rat feeding regimens and/or the lower susceptibility of these rats to EtOH. Nonetheless, these results suggest that the adverse effects of EtOH may be initially targeted to the post-meiotic events in the seminiferous tubules and/or post-testicular processes in other reproductive organs, such as the epididymis and seminal vesicle. The cauda epididymis is the site to which testicular sperm migrate and become mature. The final sperm chromatin condensation due to an increase in the disulfide bonds of protamines, which allows for compact head shape formation, occurs during sperm migration from the caput to the cauda epididymis [23]. Furthermore, caudal epididymal epithelial cells secrete proteins and small metabolites essential for sperm maturation [24,25,26,27]. Carnitine and glycerophosphocholine (GPC) are metabolites, present at a millimolar concentration in the caudal epididymal lumen, which are important for sperm metabolism, motility and survival during their epididymal residence [28,29,30,31,32,33,34]. In addition to the epididymis, the seminal vesicle is important for sperm survival and acquisition for further fertilizing competence in both male and female reproductive tracts [35,36]. It secretes fructose, an energy substrate for sperm, and citrate, a buffering component of the semen [37,38,39,40]. Although both the epididymis and seminal vesicle are androgen target tissues [41,42,43], and their normal structure and function depend on this hormone, they can also be directly affected by harmful EtOH metabolites, which would result in both structural and functional impairment. In fact, damage induced by prolonged EtOH consumption has been described in the epididymis and in the seminal vesicle. Tangsrisakda and Iamsaard [20] have described the presence of vacuoles in the epithelial cell layer of the cauda epididymis in EtOH rats as well as round cells in the epididymal lumen. In the caput epididymis of EtOH-consuming rats, Pereira et al. [44] have demonstrated a marked accumulation of lipid droplets in the principal and clear cells, whereas Sadeghzadeh et al. [45] have shown a massive increase in collagen fibers in the spaces surrounding the epididymal duct—an indication of fibrosis. In the seminal vesicle, electron microscopy indicates a decreased number of vacuoles containing secretory granules in the glandular epithelial cells, implicating the subfunctionality of the glandular secretion [46,47].

In this report, using rats as an experimental model and a prolonged EtOH consumption regimen that still allowed for ongoing spermatogenesis, we comprehensively investigated whether the cauda epididymis and seminal vesicle were selectively affected by EtOH regarding their histology, cellular viability, lipid integrity and metabolite secretion.

## 2. Materials and Methods

### 2.1. Animals

Male Wistar rats (10 weeks old; 380–400 g), purchased from Nomura Siam International the Co., Ltd. (Bangkok, Thailand), were boarded in a temperature-controlled room (23 ± 2 °C) with a photoperiod of 12 h dark/12 h light. The use and handling of rats were approved by the Institutional Animal Care and Use Committee of Khon Kaen University (protocol: IACUC-KKU-100/64), following the instructions of the Ethics of Animal Experimentation of the National Research Council of Thailand and ARRIVE checklists and guidelines.

### 2.2. Administration of Ethanol into Rats

Adult rats (n = 6) were administered 40% (*v*/*v*) ethanol (diluted from 99.9% ethanol obtained from RCI Labscan Ltd., Bangkok, Thailand) of different volumes so that they received 3 g of EtOH per kg body weight. This volume was calculated based on the density of EtOH of 0.79 g/mL. The administration was by oral gavage for 56 consecutive days, while control animals (n = 9) were administered water in parallel. The EtOH-administered rats were called EtOH animals.

### 2.3. Assessment of the Overall Behavior and Appearance of the Animals, Body Weight and Weight of Selected Organs

The overall behavior (eating/drinking and movement) and appearance of the control rats and EtOH rats was assessed daily. Both the control and EtOH animals were weighed daily until the last EtOH/water administration. Then, all animals were euthanized and their various body parts were opened for tissue collection and weighing. These tissues included reproductive organs: the testis, the epididymis and the paired seminal vesicles, and somatic organs: the brain, the paired submandibular glands, the pancreas, the stomach, the liver, the kidney, the spleen and the adrenal gland.

### 2.4. Histological Analyses of the Epididymis and Seminal Vesicle

The right side of the epididymides and seminal vesicles were rapidly fixed in 10% formalin at room temperature (RT) for 48 h, whereas the left organs were subjected to fluid collection and protein extraction. The fixed cauda epididymis and seminal vesicle were cut in the mid-transverse plane. All tissues were processed by standard methods for the preparation of paraffin-embedded tissue blocks from which sections (5 μm thick) were generated. Sections were randomly taken from five control and five EtOH rats for staining with hematoxylin and eosin (H/E; Bio-Optica, Milan, Italy), or from Masson’s trichrome kit (Abcam, Cambridge, UK) specifically detecting collagen fibers, and viewed under an Axio Imager A2 light microscope (ZEISS, Oberkochen, Germany). Microscopic images on slides were photographed with an AxioCam ICc 5 digital camera and used for morphometric analyses of the following parameters: (1) the height of the epithelial cells, (2) the distribution of collagen fibers and (3) the thickness of the smooth muscle layer surrounding the epididymal duct or the seminal vesicle wall. The measurement of these parameters was conducted in 10 different fields (2 each from each control or EtOH rat) and viewed with a 40× objective lens. At least 100 areas were measured for each parameter. The measurements and processing of the data were performed using ImageJ software (version 1.53k, National Institutes of Health, Bethesda, MD, USA). The remaining paraffin sections were used for the TUNEL assay and the immunofluorescence of caspase enzymes.

### 2.5. TUNEL Assay

For the detection of apoptotic cells on tissue sections, the TUNEL assay kit (Abcam) was used. Sections of the cauda epididymis and seminal vesicle, randomly selected from five control and five EtOH rats, were deparaffinized and rehydrated and then permeabilized with proteinase K (20 min, RT), following the manufacturer’s protocol. After washing with Tris-buffered saline (TBS, pH 7.4), the endogenous peroxidase on tissue sections was inactivated by incubating with 3% H_2_O_2_ for 5 min. Labeling reaction solutions, containing biotin-labeled deoxynucleotide triphosphate and terminal deoxynucleotidyl transferase, were gently dropped onto the sections to react with the exposed 3′OH end of the fragmented DNA strands in the apoptotic nuclei, and the sections were incubated (90 min, RT) in a moisture chamber. To probe the incorporated biotinylated nucleotides at the site of fragmented DNA, the sections were further incubated (30 min, RT) with the streptavidin-horseradish peroxidase (HRP) conjugate solution and then with diaminobenzidine solution (15 min, RT). The TUNEL-positive apoptotic cells were revealed as being of a brown color in the nuclei, and the section was counterstained with methyl green for 3 min and photographed under the Axio ImagerA2 light microscope.

### 2.6. Immunofluorescence of Caspase Enzymes

Sections of the cauda epididymis and seminal vesicle from five control and five EtOH rats were deparaffinized and rehydrated. They were then immersed in the citrate buffer (10 mM citric acid and 0.05% Tween 20, pH 6.0) and heated in a microwave at 560 Watt (5 min, three times) for antigen retrieval. After washing with phosphate-buffered saline (PBS, pH 7.4), the sections were permeabilized with 0.2% Triton X-100 (Honeywell, Inc., Charlotte, NC, USA) for 10 min in a moisture chamber. The non-specific binding of proteins was blocked with 3% bovine serum albumin (Merck, Darmstadt, Germany) in PBS for 1 h. The sections were incubated with mouse anti-caspase 9 or mouse-anti caspase 3 antibody (1:100 dilution; Santa Cruz Biotechnology, Inc., Dallas, TX, USA) overnight at 4 °C. Following washing out the unbound primary antibody with PBS, the sections were incubated (90 min) with the secondary antibody—goat anti-mouse IgG (H + L) conjugated with Alexa Flour 488 (1:300 dilution; Thermo Fisher Scientific, Inc., Waltham, MA, USA) in a dark moisture chamber. The sections were also incubated (10 min) with Hoechst 33342 (1:10,000 dilution; Abcam) to specifically stain the nuclei. All procedural steps were performed at RT unless mentioned otherwise. After mounting with glycerol, the sections were viewed under a Nikon ECLIPSE 80i epifluorescence microscope (Nikon, Tokyo, Japan) using fluorescein and Hoechst filters.

### 2.7. Processing of the Cauda Epididymis and Seminal Vesicle for Luminal Fluid Collection and Protein Extraction

These procedures were carried out on the left side of the freshly collected epididymides and seminal vesicles. Since the epididymal duct is very long and highly convoluted, with a diameter less than 1 mm, the luminal fluid had to be released during tissue mincing. Briefly, the cauda epididymis (100 mg) of the control and EtOH rats was first immersed in 1000 μL of PBS and minced into small pieces with sterile surgical scissors. This was followed by centrifugation at 14,000× *g* and 4 °C for 10 min to sediment the tissue minces. The supernatant was the diluted fluid mainly from the epididymal duct lumen and called the “caudal epididymal fluid (CEF)”, although it would also contain a small portion of fluid from the interstitial spaces. The lysate was then prepared from the sedimented epididymis tissue minces by treatment on ice with 1000 μL of RIPA buffer (Cell Signaling Technology, Inc., Danvers, MA, USA) supplemented with protease inhibitor cocktails (Merck). The treated tissue suspension was homogenized with a tissue grinder and then sonicated with an ultrasonic processor (Cole-Parmer, Vernon Hills, IL, USA) at 20 kHz for 10 times for a few seconds each time on ice. Subsequently, the sonicated suspension was centrifuged at 14,000× *g* and 4 °C for 10 min to pellet cellular particulates. The supernatant containing extracted proteins from the cauda epididymis was then stored at −20 °C until use for biochemical work.

The seminal vesicle is a sacculated glandular tissue with a lumen of a substantial volume. To collect fluid from the seminal vesicle lumen, the proximal end of the organ was held with a pair of forceps. By squeezing the tissue towards the distal end using a pair of non-tooth forceps, the luminal fluid was released into a tube containing 1000 μL of PBS. The fluid suspension was then centrifuged (14,000× *g*, 4 °C, 10 min), and the supernatant obtained was called the “seminal vesicle fluid (SVF)”. The sedimented seminal vesicle tissue was then weighed, and 100 mg was taken for further protein extraction. This was started by rinsing the fluid-voided tissue in a beaker containing PBS, followed by removing the tissue from this PBS and placing it in another tube. The tissue was then added with 1000 μL of RIPA buffer supplemented with protease inhibitor cocktails before mincing the tissue into small pieces, and the suspension was homogenized and sonicated as described above for the epididymis tissue minces. Following centrifugation (14,000× *g*, 4 °C, 10 min) to pellet cellular particulates, the supernatant containing extracted proteins from the fluid-voided seminal vesicle was stored at −20 °C until use for biochemical work.

### 2.8. Protein Quantification

Proteins in the CEF and SVF and those extracted from the cauda epididymis and seminal vesicle tissue minces were quantified on a NaNoDrop spectrophotometer (Thermo Fisher Scientific, Inc.) at A280, assuming a mass extinction coefficient (1%) of 10.

### 2.9. Immunoblotting of Caspase Enzymes

The total proteins (50 μg) extracted from the cauda epididymis and seminal vesicle from the same five control and five EtOH rats, selected for caspase immunofluorescence, were separated on 12% SDS-PAGE [48] and electrotransferred onto the nitrocellulose membrane [49]. Non-specific protein binding to the membrane was blocked with 5% skim milk dissolved in TBST (TBS containing 0.1% Tween, pH 7.4). The membrane was incubated (4 °C, overnight) with mouse anti-caspase 9 IgG or mouse anti-caspase 3 IgG antibodies (1:1000 dilution; Santa Cruz Biotechnology, Inc.) and then with the secondary antibody—rabbit anti-mouse IgG conjugated with HRP (1:5000 dilution; Merck). The membrane was also probed for the housekeeping protein, glyceraldehyde-3-phosphate dehydrogenase (GAPDH), using the primary antibody—mouse anti-GAPDH IgG (1:10,000 dilution; Abcam)—and the secondary antibody—rabbit anti-mouse IgG conjugated with HRP (1:5000 dilution; Merck). TBST was used to wash the membrane between each incubation step. The antigen–antibody complexes on the membrane were detected using an enhanced chemiluminescence (GE Healthcare, Chicago, IL, USA) system and visualized under a Gel Documentation 4 (GE Healthcare). The relative intensity of the caspase protein (both the proprotein and cleaved form) to that of GAPDH was analyzed using the ImageJ software (National Institutes of Health). The average intensity of the enzyme bands of the control rats was normalized to one. Therefore, the average expression of the enzymes in the EtOH rats was relative to the average control value of one.

### 2.10. Quantification of Malondialdehyde (MDA)

The CEF and SVF samples along with the RIPA extracts of the minced cauda epididymis and seminal vesicle were from five randomly selected control and five EtOH rats. The levels of MDA in the fluid and tissue extract were measured as previously described [21]. Briefly, the samples were mixed with thiobarbituric acid (TBA) solution containing 8.1% SDS (Bio-Rad Laboratories, Inc., Hercules, CA, USA), 0.8% TBA (Merck), and 20% acetic acid (RCI Labscan Ltd.). The mixtures were then heated (95 °C, 60 min) to allow for the formation of the pink MDA-TBA adduct. After cooling down, the mixtures were added with *n*-butanol and pyridine (15:1 dilution; RCI Labscan Ltd.) and centrifuged (4000× *g*, 10 min). The collected pink supernatant was measured for its absorbance at 543 nm. A standard curve of 1,1,3,3-tetramethoxypropane (Merck) ranging from 0 to 16 nmol/mL was constructed with an R^2^ close to 1 and used for reading MDA levels in the caudal epididymal and seminal vesicle samples. The data were expressed as ng MDA/mg of the extracted protein.

### 2.11. Nuclear Magnetic Resonance (NMR) Spectroscopic Analysis

The caudal epididymal or seminal vesicle fluids (CEF and SVF, 300 μL each) from five randomly selected control and five EtOH rats were mixed with 300 μL of NMR buffer (0.075 M Na_2_HPO_4_, 2 mM NaN_3_, 0.08% trimethylsilylpropanoic acid (TSP), pH 7.4 in D_2_O) and centrifuged (20,000× *g*, 4 °C, 15 min). The supernatant (550 μL) of each sample was transferred into a 400 MHz NMR spectrometer with CryoProbe (Bruker, Billerica, MA, USA). All ^1^H NMR spectra obtained were processed, baseline-corrected, aligned and probabilistic quotient-normalized using the MATLAB R2015a software (MathWorks, Inc., Natick, MA, USA) equipped with IMPaCTS toolbox (https://doi.org/10.5281/zenodo.3077413: accessed on 8 June 2022). Statistical total correlation spectroscopy (STOCSY) was used to verify the appearances of correlated resonances on one-dimensitional NMR spectra [50], which were searched against public databases including the human metabolome database (HMDB) and ChenomxNMR Suite version 9.0 (Chenomx, Inc., Edmonton, AB, Canada) [51,52,53,54]. To semi-quantify the metabolites in each fluid, the maximum intensity obtained from the individual peaks identified in the ^1^H NMR spectra was calculated for the concentration by referencing to the known TSP concentration [55]. A metabolic profiling experiment of all fluid samples was conducted at Khon Kaen University Phenome Center, Khon Kaen, Thailand.

### 2.12. Statistical Analysis

Except for body weight data, which were processed by two-way ANOVA with Tukey’s multiple comparison, all data were subjected to an unpaired t test (if the data were normally distributed) or a Mann–Whitney test (if the data were not distributed normally) to compare the difference between the control and EtOH-treated groups. The data normality was checked by the Shapiro–Wilk test. Statistical analyses were carried out using GraphPad Prism 9 (GraphPad Software, Inc., Boston, MA, USA). Statistically significant differences were considered when the *p* was <0.05. The data were expressed as the mean ± standard deviation (SD). For metabolomics analysis, the orthogonal partial least squares-discriminant analysis (OPLS-DA) was performed using SIMCA software version 14.1 (Umetrics, Umeå, Sweden). The quality of all OPLS-DA models was determined by R^2^ and Q^2^ values. To indicate the validity of these models, they were further assessed by the analysis of variance testing of cross-validated predictive residuals (CV-ANOVA) and considered when the permutation *p* was <0.05. S-plot analysis was then utilized to identify metabolites with different levels in EtOH and control animals. Metabolites segregated into the S-plot with a *p* value cut-off of $\left|0.05\right|$ and a *p*(corr) cut-off of $\left|0.6\right|$ were those having different levels in the EtOH animals, as compared with those in the control counterparts.

## 3. Results

### 3.1. Body Weight Growth Curve and Weight of Various Organs at the End of the EtOH Consumption Regimen

The EtOH rats showed overall growth by an increase in their body weight during the 56-day consumption regimen, with no statistical significance in their growth curve as compared with control rats. Nonetheless, the body growth of the EtOH rats showed the trend of a lower rate (Figure 1A). However, the overall appearance, eating/drinking activities and movement of the EtOH rats during the whole course of the EtOH consumption regimen did not show any difference from those of the control rats.

Among the three reproductive organs, the testis, the epididymis and the paired seminal vesicles, only the latter two organs of the EtOH rats had a statistically lower weight, as compared with those of the control counterparts, at the end of the EtOH consumption regimen (Figure 1B). These results prompted us to further investigate whether there existed histopathology, lipid peroxidation and changes in the metabolite profiles of the cauda epididymis and seminal vesicle in EtOH rats, relative to the control animals.

Of the eight somatic organs, the brain, the paired salivary glands, the pancreas, the stomach, the liver and the kidney showed no statistical differences in their weight between the EtOH and control rats on the last day of the EtOH consumption. In contrast, the weight of the spleen was decreased, while the weight of the adrenal gland was increased in the EtOH rats, as compared with the control rats (Figure 1C).

### 3.2. In Vivo EtOH Treatment Induced Histopathology of the Cauda Epididymis and Seminal Vesicle

Figure 2 and Figure 3 show the histopathology of the cauda epididymis and seminal vesicle, respectively, in EtOH rats. Sperm were present in all caudal epididymal lumen of the control rats and EtOH rats. The majority (~95%) of the sperm mass in the lumina of the EtOH rats had a similar size to that of the control rats. However, a sperm mass of a much smaller size existed in about 5% of the epididymal lumina of EtOH rats (Figure 2A). The swelling of basal cells and deletion at the apical membrane surface of the epithelial cells were observed in the cauda epididymis of the EtOH rats. Furthermore, the chromatin of a number of principal cells in the EtOH sections became condensed, and the shape of these nuclei became oblong, lying horizontally close to the basement membrane. This was in contrast to the principal cell nuclei in the control rats, the chromatin of which was not condensed, and the “rather round” nuclei were located some distance above the basement membrane (Figure 2B). Pyknotic nuclei were also observed in the epithelial cells of the EtOH rats (Figure 2B). In about 10% of the epididymal duct area of the EtOH rats, the epithelial cells became markedly delocalized, with no obvious basement membrane and a complete loss of orderliness of the epithelial cell association (Appendix A). Excluding this highly delocalized epithelium, the height of the caudal epididymal epithelium of the EtOH rats was significantly decreased when compared with that of the control rats (Figure 2C). The presence of pyknotic nuclei and the decrease in the epithelium height were also observed in the seminal vesicle of the EtOH rats, as compared with the control counterparts (Figure 3B,C). Specific to the seminal vesicle was the decrease in the mucosal folding in the EtOH rats (Figure 3A). Interestingly, another common abnormality feature in the cauda epididymis and seminal vesicle of EtOH rats was an accumulation of aniline blue-stained collagen fibers in the smooth muscle layer, which also became enlarged, as compared with that of the control counterparts (Figure 2D and Figure 3D). In the cauda epididymis, there also appeared an increase in collagen fiber clusters in the interstitial space (Figure 2D).

### 3.3. Increased MDA Levels and Apoptosis in the Cauda Epididymis and Seminal Vesicle of Rats Treated In Vivo with EtOH

Both the tissue and fluid collected from the cauda epididymis and seminal vesicle of the EtOH rats showed a significant increase in MDA levels as compared with the control counterparts (Figure 4A,B), implicating a buildup of lipid peroxidation in the EtOH animals. TUNEL analyses also indicated the presence of numerous apoptotic cells in the epithelial cell layer of the cauda epididymis and seminal vesicle of the EtOH rats (Figure 5A and Figure 6A). The TUNEL results corroborated the enhanced expression, as shown by immunofluorescence, of the two apoptotic markers, caspase 9 and caspase 3, in the cauda epididymis and seminal vesicle of the EtOH animals (Figure 5B,C and Figure 6B,C). Immunoblotting results also revealed significant increases on both the pro and cleaved forms of caspase 9 and caspase 3 in the cauda epididymal and seminal vesicle tissues in the EtOH group (Figure 5B,C and Figure 6B,C).

### 3.4. Changes in the Metabolite Composition in the Caudal Epididymal and Seminal Vesicle Fluid of the EtOH Animals

Metabolites in the CEF and SVF were shown by ^1^H NMR-based untargeted profiling using their median spectra. There were 16 metabolites in the CEF (Figure 7A) and 12 metabolites in the SVF (Figure 8A). Notably, lactate, creatine, GPC, betaine, myo-inositol and fructose were found in both CEF and SVF (Figure 7A and Figure 8A). In contrast, metabolites detected only in the CEF were alanine, acetate, acetylcarnitine, methionine, carnitine, choline, trimethylamine N-oxide, fructose 2,6-bisphosphate, adenine and formate (Figure 7A), whereas leucine, isoleucine, acetylcysteine, citrate, glycerate and xanthine were only detected in the SVF (Figure 8A). All identified metabolites in the CEF and SVF were listed in Appendix A and Appendix A, respectively. Revealed by OPLS-DA score plotting, the ^1^H NMR spectra of five replicate samples of the cauda epididymis and of the seminal vesicle in the EtOH rat group were significantly different from those of the control counterparts (R^2^X = 86.3%, Q^2^X = 0.72, permutation *p* value = 0.04; Figure 7B and R^2^X = 88.3%, Q^2^X = 0.74, permutation *p* value = 0.03; Figure 8B). The S-plots were then employed to determine the discriminatory features between the control and EtOH groups based on the critical values from the Pearson’s correlation table at a *p* value cut-off of $\left|0.05\right|$ and a *p*(corr) cut-off of $\left|0.6\right|$. The selected candidate features are colored in red, representing the metabolites with decreased levels in the EtOH group in both CEF (Appendix A) and SVF (Appendix A). Essentially, EtOH oral consumption resulted in significant decreases in the levels of alanine, carnitine, GPC, myo-inositol, fructose and fructose 2,6-bisphosphate in the CEF as compared with the corresponding levels in the control animals (Table 1). The levels of leucine, isoleucine, lactate, citrate, myo-inositol, fructose and glycerate were also significantly decreased in the SVF of EtOH rats, relative to those in the control animals (Table 1).

## 4. Discussion

In this report, we demonstrated several damage features of the cauda epididymis and seminal vesicle of rats with prolonged EtOH consumption following the regimen that still allows for ongoing spermatogenesis [20,21]. The presumption that spermatogenesis still continued in our experiment was supported by our findings that the weight of the testis of EtOH rats was not different from that of control rats (Figure 1) and the regular size of the sperm mass still existed in >95% of the caudal epididymal lumen in the tissue cross-sections of EtOH rats. The damage features in the cauda epididymis and seminal vesicle included decreases in size (weight), histopathology, increases in lipid peroxidation (revealed by elevated MDA levels) and increases in apoptosis in the two tissues, relative to the corresponding parameters in the control-untreated animals.

The histopathology in the cauda epididymis included the swelling of epididymal basal cells and the delocalization of epididymal epithelial cell nuclei, together with the reduced height of the epithelial cells. It was likely that the adverse effects of EtOH on this tissue manifested through the topical direct action of harmful EtOH metabolites rather than systemic consequences of the markedly reduced levels of male reproductive hormones (FSH, LH and androgen) [2,3,4], since, by the time that these hormones were significantly decreased, spermatogenesis would have been arrested. Nonetheless, we cannot exclude the possibility that the effect of EtOH on the spleen and adrenal gland, as seen by their weight changes in EtOH rats (Figure 1A), might have consequences on the histopathology of the epididymis and seminal vesicle. For the mechanisms involving the direct action of EtOH metabolites, ROS produced during EtOH conversion to acetaldehyde by CYP2E1 [14] would induce the peroxidation of PUFA, thus generating various lipid aldehydes including MDA (from *n*-3 PUFA) and other lipid aldehydes [11], as well as other lipid peroxidation products [11]. In fact, CYP2E1 is present in the epididymis [13], and its expression is enhanced by EtOH [11,56]. MDA, other lipid aldehydes and lipid peroxidation products would form adducts with proteins and DNA, and this could cause apoptosis and cellular damage [57]. Notably, increases in MDA levels were observed in the cauda epididymis of the EtOH rats (Figure 4). Acetaldehyde, a small lipid aldehyde, was generated from EtOH by not only the enzyme CYP2E1 [14] but also the aldehyde dehydrogenase present in the epididymis [12,58]. Acetaldehyde can also form adducts with DNA and proteins, thus adding to the induction of apoptosis (Figure 5) and cellular damage, as revealed in the histopathology (Figure 2) in the cauda epididymis of the EtOH rats.

Histopathology was also observed in the seminal vesicle of EtOH rats. This included decreases in mucosal folding and the epithelial cell height, the disorganization of some areas of the epithelial cell layer and the appearance of pyknotic nuclei (Figure 3). Although CYP2E1 has not been described in the seminal vesicle, the increase in MDA levels in this tissue of the EtOH rats suggested its existence (Figure 4), and the increase in apoptosis (Figure 6) further implicated that DNA and protein adduct formation may have occurred in the seminal vesicle of the EtOH rats, likely through mechanisms similar to those described above for the epididymis.

Interestingly, the muscle layer of both the epididymis and seminal vesicle in the EtOH rats was enlarged with an increased distribution of the collagen fibers (Figure 2 and Figure 3). There were also increased amounts of the collagen fibers in the interstitial space of the cauda epididymis in the EtOH rats (Figure 2). Our finding agrees with the previous report on the increase in the collagen fibers in the same area in the caput epididymis [45]. The abnormally higher amounts of collagen would lead to fibrosis and stiffness of the muscle layer [59] of the epididymis and seminal vesicle. Consequently, the secretion of metabolites into the lumen of these two organs may be defective due to the abnormality of the muscle contraction.

The epididymis and seminal vesicle are male reproductive organs, which support the maturation, survival and signaling/development of sperm, so they gain a fertilizing ability prior to them encountering mature ovulated eggs [24,25,26,27,30,35,36,60,61]. The metabolites secreted by these organs are important for these processes. In EtOH rats, the levels of a number of metabolites in the epididymis and seminal vesicle, important for sperm metabolism/maturation/survival, were decreased. In untreated animals, the epididymal lumen contains carnitine, GPC and myo-inositol at very high levels, as well as fructose, fructose 2,6-bisphosphate and alanine at substantial levels [38,62]. Carnitine shuttles fatty acyl chains by forming acylcarnitine through the mitochondrial membrane for β-oxidation, which generates energy [31]. Carnitine also converts acetyl-CoA into acetylcarnitine. This lowers the mitochondrial concentration of acetyl-CoA, which would otherwise inhibit pyruvate dehydrogenase, an integral enzyme in the tricarboxylic acid (TCA) cycle, and, subsequently, energy production [31]. Acetylcarnitine also directly increases the sperm motility rates [32]. GPC and myo-inositol play a role in the osmotic regulation of the epididymal fluid [33,34,63]. In addition, myo-inositol can be converted into derivatives, including phosphatidylinositol phosphate (PIP), PIP3, IP3 and IP4, all important for sperm membrane modifications, motility and signal activation [63]. Fructose present in both the epididymal and seminal vesicle lumen enters sperm to provide energy through the glycolytic pathway [37], whereas fructose 2,6-bisphosphate, a component of the epididymal fluid, enhances glycolysis through insulin action [64]. The amino acid, alanine, present in the epididymal lumen, can also be metabolized by sperm to pyruvate to enter the TCA cycle for energy production [65]. In the seminal vesicle lumen, fructose and citrate are present at very high levels. While fructose undergoes glycolysis to provide energy to sperm, as stated above, citrate gives a buffering capacity to semen for its physiological pH [38,39,40]. Glycerate and lactate in the seminal vesicle fluid are also energy-providing metabolites, with the former entering the glycolytic pathway and the latter the TCA cycle [37]. Finally, leucine and isoleucine (which could isomerize to leucine) in the seminal vesicle fluid can stimulate motility [66]. All metabolites mentioned herein were significantly decreased in the fluid from the epididymis and/or seminal vesicle in the EtOH rats, as shown by our ^1^H NMR analyses (Table 1). It is possible that these decreases could cause the observed histological abnormality of these two tissues, or vice versa. Certainly, these decreases in metabolites would lead to the improper maturation and development of sperm as well as their decreased ability to survive in EtOH males, and eventually, this would lead to male infertility/subfertility, as described previously [10,67,68,69].

In summary, our studies using rats as an experimental model suggested that initial adverse effects of EtOH after its prolonged consumption occurred at the cauda epididymis and seminal vesicle in both structural and functional aspects. Our findings herein corroborate the previous observations in EtOH rats on the decrease in caudal epididymal sperm with a normal morphology concurrent with the 50% increase in the non-viability of these sperm [20,21], and it is likely that sperm from EtOH rats have a much decreased fertilizing ability. While the amounts of alcohol administered into the rats in our study were rather high and may be equivalent to binge drinking in humans [70], our study serves as proof of the principle that EtOH orally entered into the body could exert adverse effects on the epididymis and seminal vesicle as ones of the initial targets. Further studies need to be conducted in humans drinking ethanol of various amounts. While the experimental approaches to evaluating histopathology and increases in lipid peroxidation and apoptosis of the two tissues are not practical to perform in humans, the ^1^H NMR profiling of metabolites integral to sperm development and survival can be implemented in seminal plasma prepared from ejaculated semen. Seminal plasma, together with “omics” approaches, has served as a diagnostic fluid for disorders of male reproductive physiology [71], and specifically, this NMR analysis approach has been widely used to assess human male fertility/infertility [72,73,74,75,76,77]. Since seminal plasma contains secretion from the epididymis and seminal vesicle, metabolites that come from the epididymis (such as carnitine and GPC) and from the seminal vesicle (such as fructose) should be quantified in EtOH-consuming men and compared with those in control men with no EtOH consumption. Decreased levels of these metabolites in the seminal plasma of EtOH men together with a low quality of their ejaculated sperm and a failure to impregnate their partners would be indications of impairment to their epididymis and/or seminal vesicle, which causes subfertility/infertility, and they should be advised to stop EtOH consumption. Since the continuation of EtOH consumption, especially at high amounts, will lead to damages of other organs with severe adverse effects (such as cirrhosis) in due time, the cessation of EtOH consumption following the warning from the seminal plasma metabolite data will prevent the consumers’ health from delving further into an abysmal state.

## 5. Conclusions

The prolonged administration of a high amount of EtOH into rats caused histopathology to the epididymis and seminal vesicle. Lipid peroxidation was observed in these two tissues; it could be induced by ROS, generated during the metabolic conversion of EtOH to acetaldehyde by CYP2E1. Lipid aldehydes inducing MDA and other lipid peroxidation products, as well as acetaldehyde, could form adducts with DNA and proteins, which, in turn, would result in apoptosis observed in both the epididymis and seminal vesicle. Concurrently, a number of secreted metabolites, essential for sperm maturation, metabolism and motility, in the lumen of these two tissues were decreased. Altogether, our study demonstrated that the prolonged consumption of EtOH led to structural and functional defects of the epididymis and seminal vesicle, and this was likely one of the causes of subfertility/infertility in alcohol-drinking males.

## Figures and Tables

**Figure 1 biomedicines-12-01010-f001:**
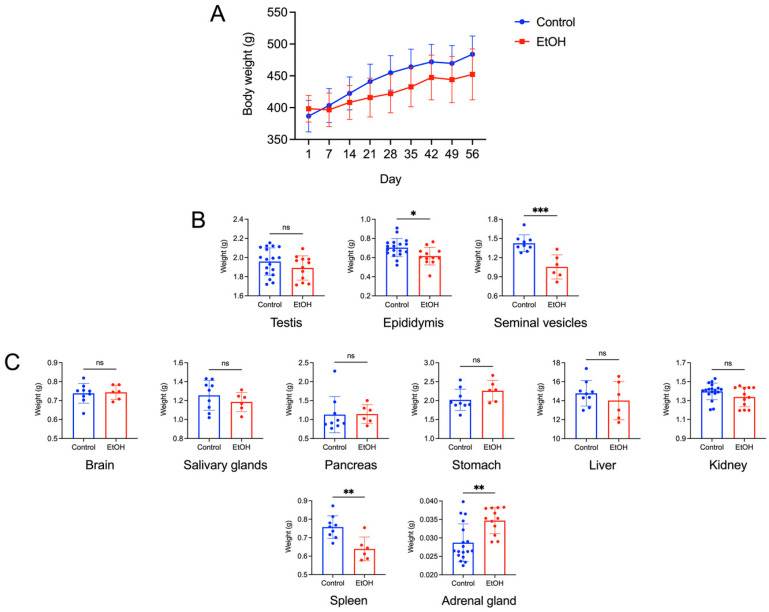
(**A**): Body weight growth curve of control and EtOH male rats during the experimental period of 56 days. The means ± SDs of the weekly body weight of each group are shown. Day 1 and Day 56 were the first and last day, respectively, of the EtOH oral administration to the EtOH rat group. The EtOH treatment regimen was 3 g of EtOH per kg of body weight daily. Water of the same weight as that of the EtOH regimen was given instead to the control animals. There were nine control (orally fed with water) rats and six EtOH rats. Two-way ANOVA showed no significant differences in the body weight increase between the control and EtOH groups during this 56-day period. (**B**): Weight of the testis, epididymis and paired seminal vesicles of control and EtOH rats. The mean ± SD of the weight of each left and right testis and of each left and right epididymis from control males (n = 18) and EtOH counterparts (n = 16) are shown in the bar graphs. However, both seminal vesicles from each animal were removed as a pair from the animal for weighing. Therefore, the mean ± SD of the weight shown was of the two seminal vesicles of each animal (n = nine and six for control and EtOH animals, respectively). Note the significant decreases in the weight of the epididymis and seminal vesicles of the EtOH rats compared with that of the control counterparts. (**C**): Weight of various somatic organs (the brain, the paired submandibular salivary glands, the pancreas, the stomach, the liver, the kidney, the spleen and the adrenal gland) of the control and EtOH rats. Each of the left and right kidneys and adrenal glands of each animal were weighed. Therefore, the number of weight values of these two organs was 18 for control animals and 12 for EtOH rats, whereas the number of values of other organs was 9 and 6 for control and EtOH rats, respectively. Note the significant differences in the weight of the spleen and the adrenal gland between the EtOH and control rats. *, ** and *** denote statistical differences with *p* < 0.05, 0.01 and 0.001, respectively.

**Figure 2 biomedicines-12-01010-f002:**
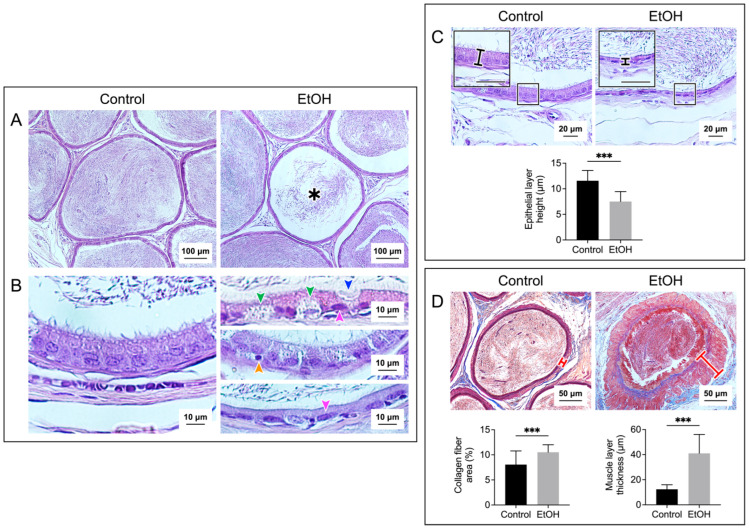
Prolonged EtOH oral administration induced histological damage to the rat epididymis. Sections of the cauda epididymis were stained with H/E in the (**A**–**C**) panels and with Masson’s trichrome in the (**D**) panel. Low-magnification images of the epididymal duct cross-sections with the sperm mass content are displayed in the (**A**) panel. A smaller sperm mass was observed in about 5% of the epididymal lumen sections of the EtOH rats (asterisk). Changes in the epididymal duct epithelium in the EtOH rats were observed at a higher magnification (**B**,**C**). Various types of epithelial cells with the well-aligned basement membrane and apical membrane were seen in the epididymal ducts of the control rats. In contrast, there was a swelling of basal cells (green arrowheads) and a deletion of the apical membrane surface of the principal cell (blue arrowhead) in the EtOH rats. In the control rats, the nearly round-shaped nuclei of principal cells were well-aligned and situated at some distance above the basement membrane, with the chromatin in the diffuse state. Contrarily, a number of the principal cell nuclei (pink arrowheads) had the chromatin in the condensed state and were flattened down right above the basement membrane ((**B**), top and bottom of the right panels). Pyknotic nuclei were also observed in some basal cells of the EtOH rats (example in (**B**), middle right panel—orange arrowhead). The height of the epididymal epithelial cells of the EtOH rats was significantly lower than that of the control cells (**C**). Using Masson’s trichrome dye to specifically stain collagen fibers (blue staining), the percentage of the collagen fiber area and the thickness of the smooth muscle area (marked by a red cross-bar) were shown to be significantly increased in EtOH rats as compared with those in the control animals (**D**). Note that there were more collagen fibers in both the muscle layer and the interstitial space of the EtOH rats. The scale bar in the insets in (**C**) is 20 μm. Note that all histology images of the cauda epididymis shown were from one control rat and one EtOH rat, although the results were representative of the five control and five EtOH rats used for the histology studies. However, the morphometric analyses were performed based on quantitative data obtained from the five control and five EtOH rats, and the data were expressed as the mean ± SD. *** denotes statistical differences with *p* < 0.001.

**Figure 3 biomedicines-12-01010-f003:**
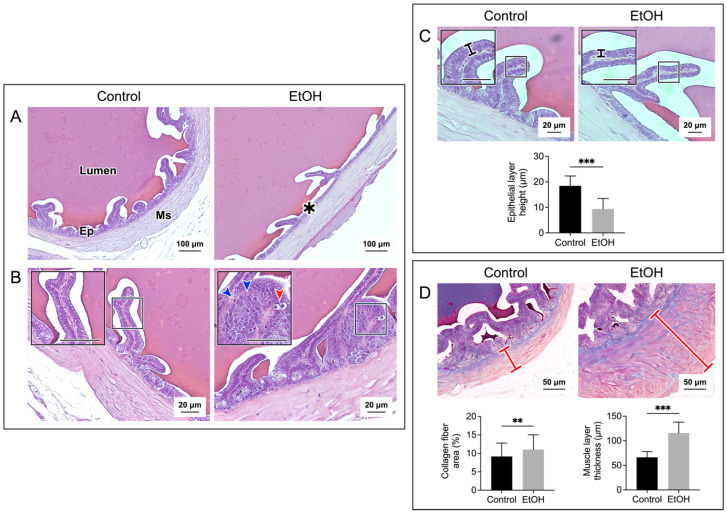
Prolonged EtOH oral administration induced histological damage to the rat seminal vesicle. Sections of the seminal vesicle were stained with H/E in the (**A**–**C**) panels and with Masson’s trichrome in the (**D**) panel. The cross-sections of the seminal vesicle containing the mucosal epithelium (Ep) and muscle layer (Ms), together with the lumen, are displayed in the (**A**) panel. Decreases in folding were observed in some areas of the mucosal epithelium of the EtOH rats (asterisk). Changes in the epithelial cell layers in the EtOH rats were observed at a higher magnification (**B**,**C**). Epithelial cells with the well-aligned basement membrane and apical membrane were seen in the control rats. In contrast, in some areas of the epithelial layer in the EtOH rats, there existed pyknotic nuclei (red arrowhead) and cell disorganization (blue arrowheads) (**B**). Overall, the seminal vesicle epithelial height of the EtOH rats was significantly decreased as compared with that of the control counterparts (**C**). Using Masson’s trichrome dye to specifically stain collagen fibers (blue staining), the percentage of the collagen fiber area and the thickness of the smooth muscle area (marked by a red cross-bar) were shown to be significantly increased in the EtOH rats as compared with those in the control animals (**D**). The scale bar in the insets is 20 μm. Note that all histology images of the seminal vesicle shown were from one control rat and one EtOH rat, although the results were representative of the five control and five EtOH rats used for the histology studies. However, the morphometric analyses were performed based on quantitative data obtained from the five control and five EtOH rats. Data were expressed as the mean ± SD. ** and *** denote statistical differences with *p* < 0.01 and 0.001, respectively.

**Figure 4 biomedicines-12-01010-f004:**
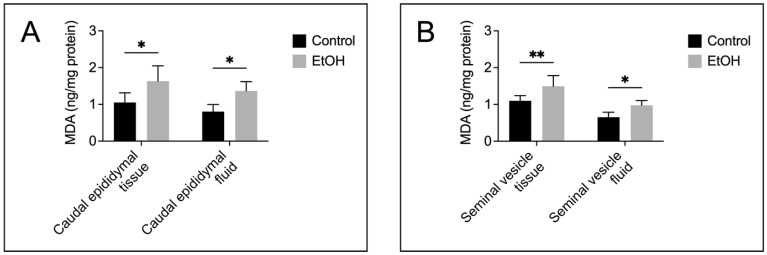
Increases in lipid peroxidation, as demonstrated by malondialdehyde (MDA) levels in the cauda epididymis (**A**) and seminal vesicle (**B**) of the EtOH rats, compared with those in the control animals. The MDA quantification was performed based on samples from five control rats and five EtOH rats. Data were expressed as the mean ± SD. * and ** denote statistical differences with *p* < 0.05 and 0.01, respectively.

**Figure 5 biomedicines-12-01010-f005:**
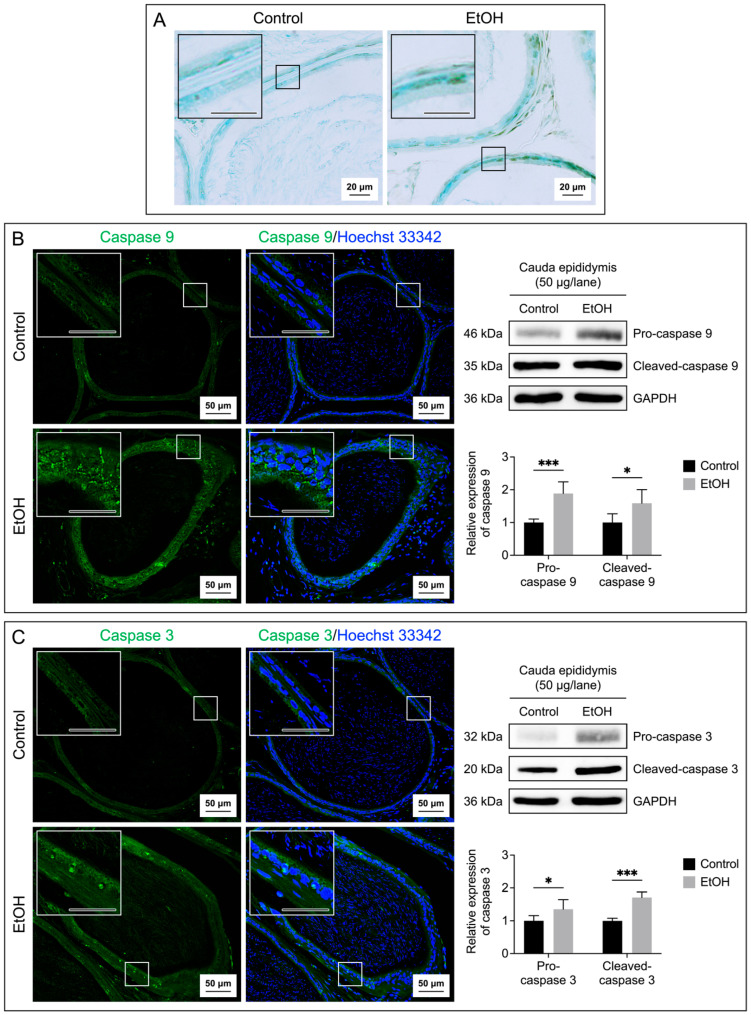
Increases in the apoptosis of the epididymal epithelial cells of the EtOH rats. Apoptosis was demonstrated by the TUNEL assay performed on the cauda epididymis sections (**A**). The brown signals indicated DNA breakages. The cauda epididymis sections were counterstained with methyl green. Immunofluorescence of caspase 9 (green signals, (**B**)) and caspase 3 (green signals, (**C**)) was also performed on the cauda epididymis sections. Nuclei of the epithelial cells were counterstained with Hoechst 33342 (blue fluorescence). The scale bar in the insets is 20 μm. Note that the TUNEL and immunofluorescence images of the cauda epididymis shown were from one control rat and one EtOH rat, although the results were representative of the five control and five EtOH rats used for these studies. Immunoblotting using anti-caspase 9 and anti-caspase 3 antibodies of the cauda epididymis tissues revealed the procaspase forms and the cleaved forms (**B**,**C**) of the enzymes, and the levels of their relative expression were quantified by densitometric analyses (bar graphs). The intensity of GAPDH was used to figure out the relative intensity of both the proenzyme and cleaved forms in each cauda epididymis sample from five control rats and five EtOH rats. The average intensity of the enzyme bands of the control rats was then normalized to one, and this normalization ratio was used to figure out the relative intensity of the procaspase forms and cleaved forms of all control and EtOH samples. The relative expression of the procaspase forms and cleaved forms was expressed as the mean ± SD. * and *** denote statistical differences with *p* < 0.05 and 0.001, respectively. Note the original immunoblotting running is shown in Appendix A.

**Figure 6 biomedicines-12-01010-f006:**
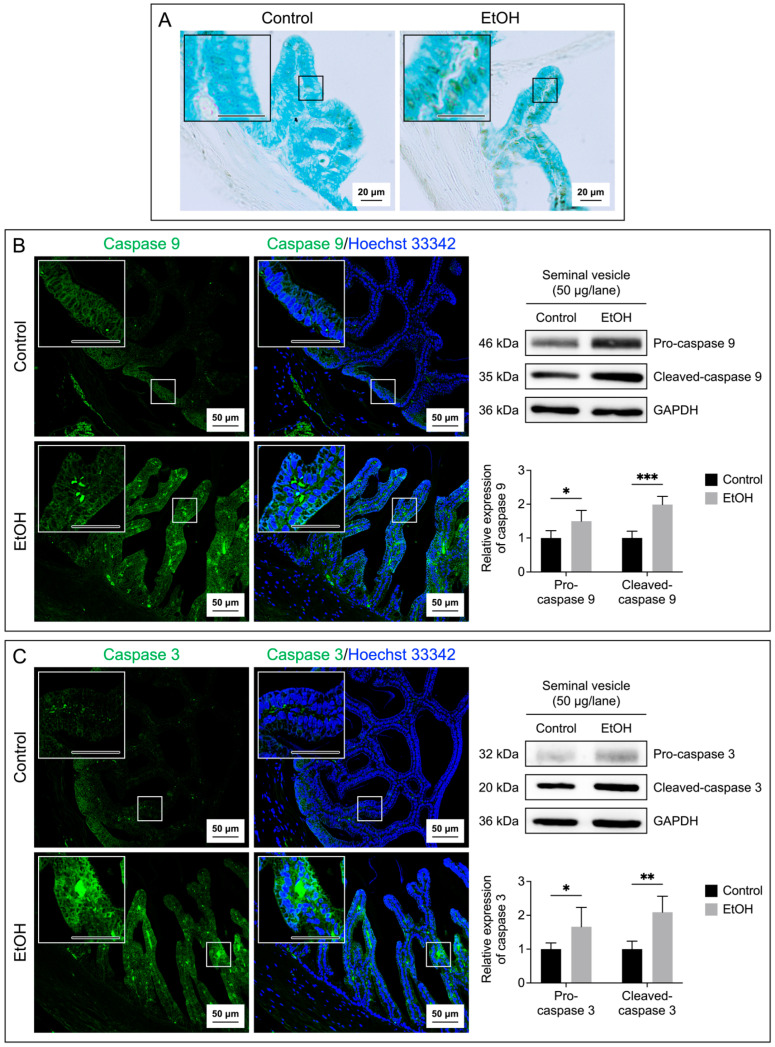
Increases in the apoptosis of the seminal vesicle epithelial cells of the EtOH rats. Apoptosis was demonstrated by the TUNEL assay performed on the seminal vesicle sections (**A**). The brown signals indicated DNA breakages. The seminal vesicle sections were counterstained with methyl green. Immunofluorescence of caspase 9 (green signals, (**B**)) and caspase 3 (green signals, (**C**)) was also performed on the seminal vesicle sections. Nuclei of the epithelial cells were counterstained with Hoechst 33342 (blue fluorescence). The scale bar in the insets is 20 μm. Note that the TUNEL and immunofluorescence images of the seminal vesicle shown were from one control rat and one EtOH rat, although the results were representative of the five control and five EtOH rats used for these studies. Immunoblotting using anti-caspase 9 and anti-caspase 3 antibodies of the seminal vesicle tissues revealed the procaspase forms and the cleaved-caspase forms (**B**,**C**) of the enzymes, and the levels of their relative expression were quantified by densitometric analyses (bar graphs). The intensity of GAPDH was used to figure out the relative intensity of both the proenzyme and cleaved forms in each seminal vesicle sample from five control rats and five EtOH rats. The average intensity of the enzyme bands of the control rats was then normalized to 1, and this normalization ratio was used to figure out the relative intensity of the procaspase forms and cleaved forms of all control and EtOH samples. The relative expression of the procaspase forms and cleaved forms was expressed as the mean ± SD. *, ** and *** denote statistical differences with *p* < 0.05, 0.01 and 0.001, respectively. Note the original immunoblotting running is shown in Appendix A.

**Figure 7 biomedicines-12-01010-f007:**
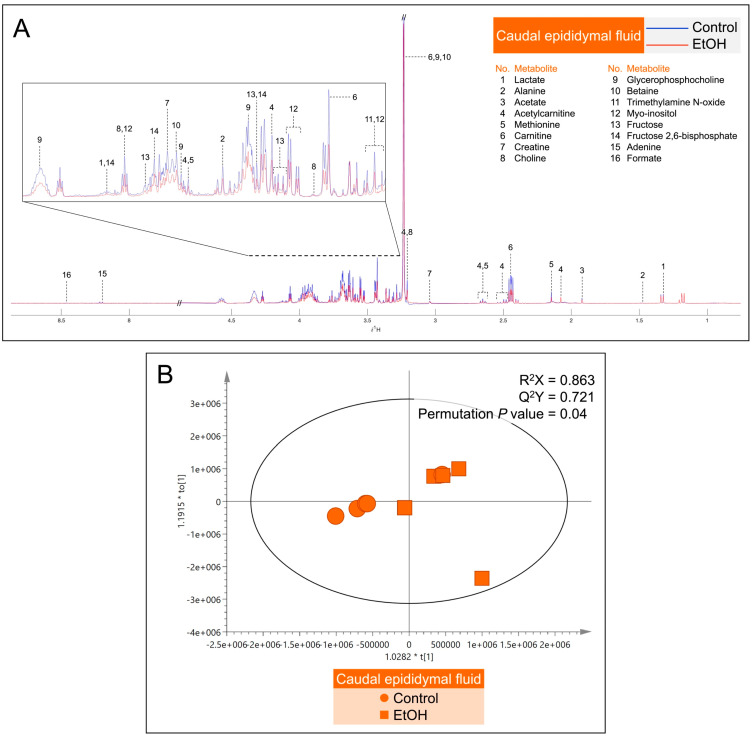
(**A**): Representative median ^1^H NMR spectra of the caudal epididymal fluid (CEF) from five control (blue peaks) rats and from five EtOH (red peaks) rats. The metabolites detected were identified, as shown in Appendix A. (**B**): Orthogonal partial least squares-discriminant analysis (OPLS-DA) score plot of the ^1^H NMR spectral data of CEF metabolites from five control rats and five EtOH rats. Note that the data of two EtOH samples were almost 100% overlapping, making the plot of the EtOH group appear as having only four points. A permutation *p* value = 0.04 indicates the significant differences in metabolites between the control and EtOH rats.

**Figure 8 biomedicines-12-01010-f008:**
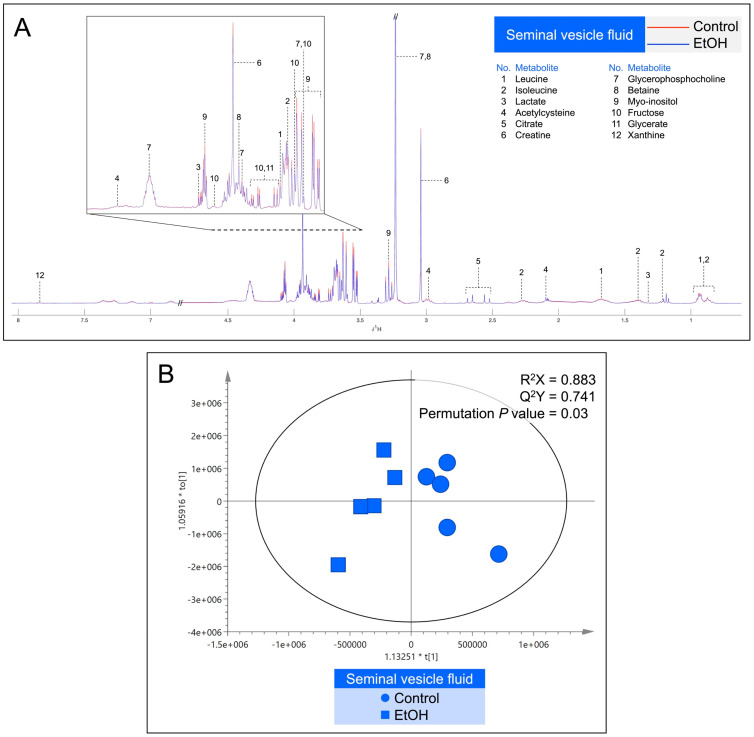
(**A**): Representative median ^1^H NMR spectra of the seminal vesicle fluid (SVF) from five control (red peaks) rats and from five EtOH (blue peaks) rats. The metabolites detected were identified, as shown in Appendix A. (**B**): Orthogonal partial least squares-discriminant analysis (OPLS-DA) score plot of the ^1^H NMR spectral data of metabolites from SVF metabolites from five control rats and five EtOH rats. A permutation *p* value = 0.03 indicates the significant differences in metabolites between the control and EtOH rats.

**Table 1 biomedicines-12-01010-t001:** Significant changes in metabolite levels in the fluid of the cauda epididymis (CEF) and seminal vesicle (SVF) in the EtOH rats as compared with those in the control animals.

	Control (mM) ^#^	EtOH (mM) ^#^
**Caudal epididymal fluid (CEF)**		
Alanine	0.92 ± 0.23	0.41 ± 0.16 *
Carnitine	13.66 ± 3.15	6.44 ± 2.61 *
Glycerophosphocholine (GPC)	4.55 ± 1.05	2.15 ± 0.87 *
Myo-inositol	0.90 ± 0.33	0.38 ± 0.17 *
Fructose	1.00 ± 0.27	0.43 ± 0.34 *
Fructose 2,6-bisphosphate	0.72 ± 0.13	0.37 ± 0.16 *
**Seminal vesicle fluid (SVF)**		
Leucine	3.98 ± 0.37	2.58 ± 0.64 *
Isoleucine	0.85 ± 0.12	0.61 ± 0.17 *
Lactate	2.55 ± 0.21	1.87 ± 0.41 *
Citrate	2.19 ± 0.72	1.45 ± 0.28 *
Myo-inositol	6.10 ± 0.65	4.83 ± 0.81 *
Fructose	3.01 ± 0.37	2.19 ± 0.42 *
Glycerate	3.04 ± 0.43	2.19 ± 0.46 *

^#^ Data are presented as the mean ± SD in mM from five control animals and five EtOH rats. * denotes statistical differences with *p* < 0.05.

## Data Availability

All data from this study are contained within this article and Appendix A. Additional details of these data will be shared in response to a reasonable request to the corresponding authors.

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
