# Peer review of "Rats Orally Administered with Ethyl Alcohol for a Prolonged Time Show Histopathology of the Epididymis and Seminal Vesicle Together with Changes in the Luminal Metabolite Composition"

_biomedicines, 2024, doi:10.3390/biomedicines12051010_

Round 1
Reviewer 1 Report
Comments and Suggestions for Authors
The manuscript is interesting but requires major changes before it is considered for publication.
The introduction could be strengthened by incorporating a few comments that highlight the relation of metabolites to spermatogenesis.
The major flaw is the metabolomics part.
The methods section must be rewritten. Were the spectra aligned? and how?
Identification of metabolites cannot be performed with STOCSY, the latter method enables the identification of features responsible for the metabolite variation.
Were 2D experiments implemented? how did the authors ascertain the determination of these metabolites?
The R2 and Q2 values present the quality of the models, the validity must be presented through ROC curves and permutation testing with 999 permutations.
The authors used OPLS-DA models and no S-line plots. These plots should be presented to verify the metabolite trends and reveal which metabolites contribute most to the discrimination otherwise the results and discussion part are in doubt.
Did the authors facilitate enrichment or pathway analysis?
Comments on the Quality of English LanguageMinor editing of English language required
Reviewer 2 Report
Comments and Suggestions for Authors
The authors provide plausible research into the link between alcohol consumption and male infertility. The research is of high quality. The figures and descriptions provide replicability of the study.
The following revisions should be considered:
- the title could be improved to better specify the contents of the paper. For example, it is not quite clear that this is an in vivo study in rats
- Lines 41, 44 and throughout: the stigmatizing terms "alcoholic" and "alcoholism" should be avoided
- Line 115: "of a various volume" is unclear. Do you mean variable volume?
- Section 2.2. and discussion: can you provide a remark about the exposure level and its relevancy for humans drinking alcohol. I think for humans the level of 3 g/kg bw would be quite, if not impossibly high? Therefore, how relevant is the study for humans?
- line 275: i would use non-normal not abnormal
Reviewer 3 Report
Comments and Suggestions for Authors
The submitted manuscript is of a very high quality. Shortly speaking, it describes the results of administration of ethanol on the reproductive system of male rats. The topic is important from both scientific and practical points of view.
I have only some minor suggestions:
Line 20, instead of „fed” it should be “administrated”
Line 36, why “have been” and not “are”?
Line 106, how many rats have been used?
Line 115, why this particular amount has been chosen?
Line 116, at what time was the administration? And was it in one dose?
What was the origin of alcohol? Who was the producent? Was it 98% alcohol?
Lines 254-255, I guest that the deuterated solvents have been used?
Line 273, what test has been used to check normality?
Were the NMR signals somehow deconvoluted?
Line 576, here, the separate “Conclusions” section should be created
Round 2
Reviewer 1 Report
Comments and Suggestions for Authors
I am very content with the effort the authors exhibited with their answers and with the new improved version of the manuscript. I believe that it is fit for publication.
Comments on the Quality of English LanguageThe quality of the English is fine